# Sorafenib Resistance Contributed by IL7 and MAL2 in Hepatocellular Carcinoma Can Be Overcome by Autophagy-Inducing Stapled Peptides

**DOI:** 10.3390/cancers15215280

**Published:** 2023-11-03

**Authors:** Jeffrey C. To, Shan Gao, Xiao-Xiao Li, Yanxiang Zhao, Vincent W. Keng

**Affiliations:** 1Shenzhen Research Institute, The Hong Kong Polytechnic University, Shenzhen 518057, China; jeffrey-cw.to@connect.polyu.hk (J.C.T.); xiaox.li@polyu.edu.hk (X.-X.L.); 2State Key Laboratory of Chemical Biology and Drug Discovery, Department of Applied Biology and Chemical Technology, The Hong Kong Polytechnic University, Kowloon, Hong Kong, China; 3Princess Margaret Cancer Centre, University Health Network, Toronto, ON M5G 1L7, Canada

**Keywords:** hepatocellular carcinoma, sorafenib resistance, IL7, MAL2, survival signaling pathways, autophagy

## Abstract

**Simple Summary:**

Drug resistance remains a major challenge in the treatment of hepatocellular carcinoma (HCC). Our objective was to investigate the genetic mechanisms involved in the development of Sorafenib-associated drug resistance in HCC cells. To achieve this, transposon insertional mutagenesis was utilized as a forward genetic tool to generate Sorafenib-resistant HCC cell lines and identify potential drug resistant genes. Our studies identified two genes, interleukin 7 (*IL7*) and mal, T cell differentiation protein 2 (*MAL2*), which conferred resistance to Sorafenib by activating the pro-survival JAK/STAT and PI3K/AKT signaling pathways. Additionally, autophagy inducing stapled peptides exhibited an anti-proliferative effect on both wild-type and drug-resistant HCC cells. When combined with Sorafenib, these peptides demonstrated efficacy against drug-resistant cells. Therefore, targeting autophagy could present a novel approach for overcoming Sorafenib-associated drug resistance in HCC induced by *IL7*- and *MAL2*-overexpression.

**Abstract:**

Drug resistance poses a great challenge in systemic therapy for hepatocellular carcinoma (HCC). However, the underlying molecular mechanisms associated with resistance to anti-cancer drugs, such as Sorafenib, remain unclear. In this study, we use transposon insertional mutagenesis to generate Sorafenib-resistant HCC cell lines in order to identify potential drug resistant causative genes. Interleukin 7 (*IL7*) and mal, T cell differentiation protein 2 (*MAL2*) were identified as candidate genes that promote survival by activating JAK/STAT and PI3K/AKT signaling pathways. Sorafenib-resistant cells exhibited higher clonogenic survival and lower drug sensitivity due to *IL7* and *MAL2* upregulation. Higher anti-apoptotic effect, clonogenic survival and increased PI3K/AKT/STAT3 activities were observed in *IL7* and *MAL2* co-overexpressing cells compared with controls or cells overexpressing *IL7* or *MAL2* individually. Given the critical role of MAL2 in endocytosis, we propose that MAL2 might facilitate the endocytic trafficking of IL7 and its cognate receptors to the plasma membrane, which leads to upregulated JAK/STAT and PI3K/AKT signaling pathways and Sorafenib resistance. Additionally, our previous studies showed that an autophagy-inducing stapled peptide promoted the endolysosomal degradation of c-MET oncogene and overcame adaptive Sorafenib resistance in c-MET^+^ HCC cells. In this study, we demonstrate that these stapled peptides readily induced autophagy and inhibited the proliferation of both wild-type and Sorafenib-resistant HCC cells co-overexpressing both *IL7* and *MAL2*. Furthermore, these peptides showed synergistic cytotoxicity with Sorafenib in drug-resistant HCC cells co-overexpressing both *IL7* and *MAL2*. Our studies suggest that targeting autophagy may be a novel strategy to overcome IL7/MAL2-mediated Sorafenib resistance in HCC.

## 1. Introduction

Although human hepatocellular carcinoma (HCC), i.e., liver cancer, is the sixth most commonly diagnosed cancer and the fourth leading cause of cancer death worldwide, there are only a few Food and Drug Administration (FDA)-approved drugs for the treatment of this disease. To ensure better treatment regimens and prognosis of HCC, it is imperative to detect the disease as early as possible. At early stages of the disease, surgical removal of the tumor is the standard therapy for HCC patients with well-preserved liver functions [1]. As a result of this early intervention, the majority of these early-stage HCC patients have a relatively high five-year survival rate [1,2]. However, late detection of the disease and the limited availability of suitable donors for liver transplantation can greatly affect treatment efficiency in advanced stage patients [1]. Unfortunately, the majority of HCC patients are diagnosed at an advanced state, with chemotherapy as their only treatment option [1]. To date, there are four kinase inhibitors that are approved by the FDA for the treatment of advanced-stage HCC: Sorafenib, Lenvatinib, Regorafenib and Cabozantinib [3]. Recently, two combinational immunotherapy regimens have also been introduced: Nivolumab and Ipilimumab [4,5]; Atezolizumab and Bevacizumab [6,7].

Sorafenib targets various multiple intracellular and cell surface kinases, such as Raf-1 proto-oncogene, serine/threonine kinase (RAF1), vascular endothelial growth factor receptors (VEGFR1, VEGFR2 and VEGFR3), and platelet derived growth factor receptor beta (PDGFRB), to inhibit angiogenesis and induce apoptosis [8]. In a clinical trial, 602 advanced HCC patients received either Sorafenib or placebo, and the results demonstrated that the Sorafenib-treated group exhibited longer median overall survival (OS) than the placebo group (10.7 vs. 7.9 months; *p* < 0.001) and longer median time to progression (TTP) (5.5 vs. 2.8 months). After 1 year, 44% of patients who received Sorafenib were still alive, while 33% of patients who received the placebo were still alive [9]. Hand–foot skin reaction (HFSR), weight loss, diarrhea and hypophosphatemia were reported frequently in the Sorafenib-treated group [9]. Based on these results, Sorafenib was approved in 2007 and was the earliest drug approved by FDA, becoming the standard first-line systemic treatment for patients with advanced HCC. Unfortunately, Sorafenib can only modestly extend the overall survival of advanced-stage HCC patients. In addition, these patients usually experience reoccurrence due to primary and acquired drug-resistances, limiting the beneficial effects of the drugs [3].

Traditionally, it is extremely challenging and labor-intensive to generate drug resistance models in vitro that can identify contributing genes based on differential gene expression profiling. In light of this, a novel method using Sleeping Beauty (*SB*) transposon insertional mutagenesis was used to generate drug-resistant human liver cancer cells in order to identify potential drug resistant causative genes. The *SB* transposon insertional mutagenesis system generates gain-of-function, loss-of-function or transposon-endogenous gene fusions [10,11,12]. This method could be used to provide important clues for identifying genes that contribute to drug-resistance; it is a traceable and effective method to rapidly generate various resistant cell line models using high dosage drug selection after *SB* transposon mutagenesis.

Using this approach, interleukin 7 (*IL7*) and mal, T cell differentiation protein 2 (*MAL2*) were identified as contributing to HCC-associated Sorafenib resistance by dysregulating the JAK/STAT and PI3K/AKT signaling pathways. *IL7* has been demonstrated to play a critical role in both B-cell and T-cell proliferation and immature immune cell arrest occurs in cells lacking *IL7* [13,14]. The binding of IL7 and IL7 receptor on the cell membrane activates Janus kinase family members (JAK1 and JAK3) and upregulation of various downstream signaling pathways, such as the signal transducer and activator of transcription (STAT) protein family and PI3K, resulting in proliferation and anti-apoptosis [13,15,16]. MAL2 is a transmembrane protein belonging to the MAL proteolipid family and contributes the cellular functions of transcytosis, secretion and the endocytic transport pathway [17]. In this study, Sorafenib-resistant cells exhibited higher clonogenic survival and lower drug sensitivity due to *IL7* and *MAL2* upregulation. Higher anti-apoptotic effect, clonogenic survival and increased PI3K/AKT/STAT3 activities were observed in *IL7* and *MAL2* co-overexpressing cells compared with controls or cells individually transfected with *IL7* or *MAL2* overexpressing plasmids. Therefore, it is hypothesized that MAL2 might facilitate trafficking of IL7 and its cognate receptors to the plasma membrane and contribute to HCC-associated Sorafenib resistance by upregulating important survival signaling pathways. Additionally, our previous studies showed that an autophagy-inducing stapled peptide promoted the endolysosomal degradation of the MET proto-oncogene, receptor tyrosine kinase (MET) oncogene and overcame adaptive Sorafenib resistance in MET^+^ HCC cells. In this study, we demonstrate that these stapled peptides readily induced autophagy and showed synergistic cytotoxicity with Sorafenib in drug-resistant HCC cells co-overexpressing both *IL7* and *MAL2*. Our studies suggest that targeting autophagy may be a novel strategy to overcome IL7/MAL2-mediated Sorafenib resistance in HCC.

## 2. Materials and Methods

### 2.1. Cell Culture

The PLC/PRF/5 human liver cancer cell line from the ATCC Liver Cancer Panel (TCP-1011) was used in this study. PLC/PRF/5 cells were cultured in Dulbecco’s modified Eagle’s Medium (DMEM), supplemented with 10% fetal bovine serum (FBS) and 1% Antibiotic-Antimycotic (Anti-Anti), in a humidified 5% CO_2_ incubator at 37 °C. All cell culture media and reagents were purchased from Life Technologies, Carlsbad, CA, USA.

### 2.2. Determining the Various Inhibitory Concentrations of Sorafenib in PLC/PRF/5 Cells

Sorafenib (MedChem Express, Monmouth Junction, NJ, USA) was used from 0 to 25 µM to determine the inhibitory concentrations (IC) in a 5-day kill curve assay. Based on the assay, the IC_20_, IC_50_ and IC_90_ for parental PLC/PRF/5 cells were 2 µM, 5.5 µM and 8 µM, respectively.

### 2.3. Establishing Sorafenib-Resistant Cells Using the Sleeping Beauty Transposon Insertional Mutagenesis System

The *SB* transposon system was used as a gene delivery tool for long-term expression delivery and integration into the host chromosomal DNA. The system is composed of two components: transposon containing the gene(s)-of-interest flanked by inverted repeat/direct repeats (IR/DR) sequences; and the other is the transposase, enzyme responsible for binding the IR/DRs, excising and randomly integrating the transposon into the genome at TA-dinucleotide sites [18]. The *SB* transposon mutagenesis system is similar to the above but relies on the mutagenic transposon called T2/Onc3 (a gift from Dr Adam J. Dupuy, Department of Anatomy and Cell Biology, University of Iowa, Iowa City, IA, USA) to generate gain-of-function and loss-of-function mutations as the result of its integration into the host chromosome [10,19,20]. The T2/Onc3 consists of splicing acceptors (SA) with a polyadenylation sequence to interrupt endogenous splicing mechanisms, resulting in loss-of-function [10]. It also consists of a CMV enhancer/chicken β-actin promoter (CAG) with splice donor (SD), which is a strong promoter enhancer element that can cause misexpression or gene truncations when integrated into the host genome, causing a gain-of-function mutation [10]. The hyperactive SB100 transposase was used in this study [21]. Transfection was performed using 3 μg of T2/Onc3 transposon plasmid, 1 μg of SB100 transposase plasmid and 8 μL of ViaFect™ Transfection Reagent (Promega, Madison, WI, USA) per 1 × 10^5^ cells. After 1 week of exposure to *SB* transposons mutagenesis, cells with 8 μM Sorafenib (MedChem Express) for PLC/PRF/5 cells were selected, until all parental cells were nearly dead (~1 week). Cells that survived the selection process were treated with 2 μM Sorafenib to allow for recovery and growth to 80% confluence, where they were maintained by culturing in the presence of Sorafenib at their parental IC_50_ concentration.

### 2.4. Chemosensitivity Assay

Chemosensitivity was measured by 3-(4,5-dimethylthiazol-2-yl)-5-(3-carboxymethoxyphenyl)-2-(4-sulfophenyl)-2H-tetrazolium (MTS) assay (Promega) using the manufacturer’s protocols. Briefly, Sorafenib-resistant cells were cultured at a cell density of 5000 in 96-well plates and treated with Sorafenib at various concentrations. After 72 h, 10 μL of MTS solution was added to each well. After incubation for 1.5 h, absorbance was measured at 452 nm using a Ledetect96 microplate reader (Labexim Products, Lengau, Austria).

### 2.5. Clonogenic Survival Assay

Cells were seeded at a density of 5000 per well in a 6-well plate cultured with media containing Sorafenib (5 or 10 μM) or peptides for 10 to 12 days. The wells were then washed twice with PBS, fixed with methanol and then stained with 0.1% crystal violet. Subsequently, the plates were scanned for semi-quantification using the ImageJ software 1.51j8 (NIH, Bethesda, MD, USA).

### 2.6. RNA-Sequencing (RNA-Seq)

Beijing Genomics Institute (BGI) was outsourced to perform the RNA-seq and bioinformatics analyses to determine differentially expressed genes (DEGs) that are statistically significant between the wild-type parental and Sorafenib-resistant cell lines. Sequencing and expression level of each gene was calculated by fragments per kilobase of exon per million reads mapped (FPKM). Based on the gene expression level, differentially expressed genes (DEGs) were identified between parental and drug-resistant cell lines. Fold-change in gene expression between two samples was calculated by log2 FPKM ratio of two samples.

### 2.7. Construction of IL7 and MAL2 Overexpression Plasmids for Transfection into Human Hepatic Cell Line

The piggyBac (*PB*) transposon system was used to stably integrate and overexpress *IL7* and *MAL2* into the hepatic genome of PLC/PRF/5. pHAGE-CMV-hIL7-IRES-ZsGreen-W was a gift from Dr David Baltimore (Addgene plasmid #26532; http://n2t.net/addgene:26532 accessed on 30 October 2019; RRID: Addgene_26532) [22] and was used as cDNA template of *IL7* for high fidelity PCR using primer pairs *Kpn*I-Kozak-*IL7* forward CAGGGTACCGCCACCATGTTCCATGTTTCTTTTAG (5′ to 3′) and *Eco*RV-*IL7* reverse CGTACGATATCTTAGTGTTCTTTAGTGCCCA (5′ to 3′); it was then inserted into the pENTR entry vector (Thermo Fisher Scientific, Waltham, MA, USA) using specific restriction enzymes, resulting in pENTR-*IL7*. The expression vector was obtained by introducing the *IL7* from the entry clone to the destination vector, pPB/SB-DEST-GFP-Puro [23], to yield pPB/SB-*IL7*-GFP-Puro. The Lenti ORF clone of human *MAL2*, Myc-DDK-tagged (OriGene Technologies; RC203862L1) was also used as cDNA template of *MAL2* for high fidelity PCR using primer pairs *Kpn*I-Kozak-*MAL2* forward CAGGGTACCGCCACCATGTCGGCCGGCGGAGCGTC (5′ to 3′) and *Eco*RV-*MAL2* reverse CGTACGATATCTTACGGTCG CCATCTTCGTA (5′ to 3′), then cloned into a *PB* transposon expression vector using a similar method for generating pPB/SB-*IL7*-GFP-Puro. A control expression vector containing the green fluorescent protein (GFP) was generated using the similar method to the one that yielded pPB/SB-GFP-Puro. These expression vectors were then co-transfected with the *PB* transposase vector into human HCC cells using the ViaFect transfection reagent (Promega) according to the manufacturer’s instruction. Puromycin (Thermo Fisher) selection post-transfection was used for enrichment and to achieve stably transfected cells.

### 2.8. Apoptosis Assay

After 10 μM Sorafenib treatment for 3 days, transfected cells were stained by APC-conjugated Annexin V and propidium iodide (PI) in Annexin V binding buffer (BioLegend, San Diego, CA, USA) at room temperature for 15 min. Apoptosis percentage was determined using BD Accuri C6 flow cytometer (University Research Facility in Life Sciences of The Hong Kong Polytechnic University, Hong Kong). Flow cytometry profiles represented intensity of Annexin-V-APC staining in *X*-axis and PI in *Y*-axis.

### 2.9. Real-Time Quantitative Reverse Transcription-Polymerase Chain Reaction (qPCR)

Template cDNA for qPCR was synthesized from 500 ng of total RNA that was extracted from transfected cells using PrimeScript RT Master Mix (Takara, Bio Inc., Kusatsu, Japan) according to the manufacturer’s protocol. The cDNA was diluted 1:10 using nuclease-free water, and 4 µL of the diluted cDNA was used to perform qPCR using SYBR Green I containing qPCR master mix (GoTaq qPCR Master Mix, Promega) with specific primers (0.2 µM final concentration of each primer). Reactions were run on a QuantStudio7 Flex Real-time PCR System (Thermo Fisher) (University Research Facility in Life Sciences of The Hong Kong Polytechnic University, Hong Kong). Primer sequences were as follows: *ACTB* forward 5′-GCCGTCTTCCCCTCCA TCGT-3′ and reverse 5′-TGCTCTGGGCCTCGTCGC-3′; *IL7* forward 5′-GACAGCATGAA AGAAATTGGTAGC-3′ and reverse 5′-CAACTTGCGAGCAGCACGGAAT-3′; *MAL2* forward 5′-GCCACATCCCTGCATGATTTGC-3′ and reverse 5′-CGTAAAGCCAGACCC AAACTGC-3′; *CDKN1A* forward 5′-AGGTGGACCTGGAGACTCTCAG-3′ and reverse 5′-TCCTCTTGGAGAAGATCAGCCG-3′.

### 2.10. Western Blot Analyses

Protein was extracted from transfected cells using a standard SDS protein lysis buffer. Protein concentrations were determined using standard protein assay (Bio-Rad, Hercules, CA, USA) with 10 µg of protein loadings separated by SDS-PAGE and transferred onto PVDF membranes. Primary antibodies STAT3 (Abcam, Cambridge, MA, USA), p-STAT3 (Abcam), PI3K (Cell Signaling Technology, CST, Danvers, MA, USA), p-PI3K (CST), AKT (CST), p-AKT (CST), JNK (Santa Cruz Biotechnology, SCB, Texas, CA, USA), p-JNK (SCB) and ACTB (ABclonal, Woburn, MA, USA) were diluted in 1% BSA at 1:1000 concentrations. Secondary anti-mouse or anti-rabbit antibodies (CST) were diluted in 1% BSA at 1:2000 concentrations. Membranes were blocked with 5% non-fat milk, then incubated with a primary antibody at 4 °C overnight followed by secondary antibody incubation at room temperature for an hour. The membranes were washed 3 times with 1X TBST at 15 min intervals. The membranes were visualized using Immobilon Western Chemiluminescent HRP Substrate (Bio-Rad) and then exposed by ChemiDoc Touch Imaging System (Bio-Rad). Semi-quantitative analyses of protein bands were measured using the NIH ImageJ software. The intensity of bands was calculated as an arbitrary value relative to ACTB expression level.

### 2.11. Semi-Quantitative Analyses

ImageJ software 1.51j8 (NIH) was used to perform semi-quantitative analyses of clonogenic assays, non-saturated RT-qPCR amplicons and Western blot bands. Briefly, the intensity of non-saturated RT-qPCR amplicon bands was measured as an arbitrary value relative to *ACTB* expression levels, while intensity of probed bands from Western blot was measured as an arbitrary value relative to ACTB protein levels.

### 2.12. Immunoblot Analysis for Autophagy

Cells were plated in a 6-well plate at a final confluence of 60%. They were then treated with rapamycin or peptides for the specified duration. After treatment with rapamycin (500 nm, 3 h) or Chloroquine (1 μM, 3 h), Tat-SP9 and Tat-SP4 cells were washed with PBS and lysed using Laemmli sample buffer (62.5 mM Tris-HCl, pH 6.8, 2% SDS, 25% glycerol, and 5% β-mercaptoethanol) supplemented with EDTA-free protease inhibitor cocktail. The protein samples obtained were separated by SDS-PAGE gel and transferred to a PVDF membrane (Millipore, Burlington, MA, USA). These membranes were incubated with anti-LC3 antibody (Abnova, Bedford, MA, USA), anti-p62 antibody (Abnova, Bedford, MA, USA) and β-actin (Santa Cruz Biotechnology, SCB, Texas, CA, USA), followed by HRP-conjugated secondary antibodies. Protein bands on the membrane were visualized using ECL reagents and analyzed using ImageJ.

### 2.13. Cell Viability Assay Using Trypan Blue Dye

Cell viability was evaluated through the trypan blue exclusion assay. The cells were seeded in 96-well plates and incubated overnight before being treated with peptides. Tat-SP9 and Tat-SP4 were administered at varying concentrations for 24 h. The Z1 Particle Counter (Beckman Coulter, Roissy, France) was used to determine the number of viable cells, and IC_50_ values were calculated by fitting concentration–response data sets to curves.

### 2.14. Chemical Synthesis of Stapled Peptides

The Tat-SP4 and Tat-SP9 were purchased from GL Biochem (Shanghai, China) Ltd., who handled the synthesis, purification and characterization. The sequence information and synthesis process are identical to our previous studies [24,25]. The full sequence of Tat-SP4 and Tat-SP9 are Ac-[Tat]-RLISEL(R8)DREKQR(S5)A-NH2 and Ac-[Tat]-VLFN(R8)LVDVIK (S5)R KV-NH2, respectively. In brief, the peptides were synthesized using an automated solid-phase technique. Olefin-containing amino acids were incorporated at specific positions following the chemical synthesis method pioneered by Verdine et al. [26,27]. The hydrocarbon staple was introduced on these amino acids through a ring-closing metathesis reaction with the Grubbs catalyst. The importance of the hydrocarbon staple in maintaining the α-helical structure of the designed peptides is confirmed by circular dichroism (CD) measurements. Chemical structure and purity of the final product were characterized by HRMS and HPLC as shown in our previous paper [24,25]. The purity of each peptide is >95%. Stock solution of the peptide was prepared by dissolving the sample in pure water to a concentration of 20 mM and stored at −20 °C.

### 2.15. Statistical Analyses

Values are presented as mean ± SEM. Statistical significance was assessed by two-tailed, unpaired Student’s *t*-test or two-way ANOVA test (Prism Software, version 7.0). *p* Values < 0.05 were considered as statistically significant.

## 3. Results

### 3.1. SB Transposon Insertional Mutagenesis Induced Drug Resistance in Human HCC Cell Line

Five-day kill curves using different concentrations of Sorafenib were performed to measure the different IC values for the use of drug-resistance selection in PLC/PRF/5 cells (Figure 1A). The *SB* transposon system utilized in this screening relies on the excision and random integration of mutagenic transposons into the hepatic genome by transposase to either enhance or disrupt gene expression (Figure 1B). For the transposon-induced mutagenesis stage, PLC/PRF/5 parental cells were transfected with both *SB* transposon (T2/Onc3) and transposase (SB100) vectors in a 3:1 ratio, respectively. Resulting *SB*-mutated cells that were able to survive/grow in the presence of Sorafenib at their parental IC_50_ concentration for maintenance state after selection and recovery states were isolated (Figure 1C). The *EGFP* fluorescent signal expressed from the SB100 vector was used as an indication of the transfection efficacy. Since ~80% *EGFP* fluorescent signal was detected post-transfection, it was assumed that most cells were successfully transfected with both the T2/Onc3 and SB100 vectors. To confirm for successful transposon insertions in transfected PLC/PRF/5 cells, their total RNA was extracted and revered to cDNA for PCR analyses to amplify the splice acceptor (SA) sequence found in the transposon vector (Figure 1B). PCR was successfully conducted to amplify a 151 bp fragment in PLC/PRF/5^Sor-R^ mutant pool 1, confirming the insertion of the mutagenic transposon (Figure 1D).

The *SB* transposon insertional mutagenesis system can generate gain-of-function, loss-of-function or transposon-endogenous gene fusions. It is hypothesized that some cells undergoing transposon mutagenesis will gain the capability of becoming Sorafenib-resistant. Therefore, the five-day minimum fatal concentration of Sorafenib was measured to remove non-targeted cells during the selection process (Figure 1C). Sorafenib concentration of 8 μM (IC_90_ value) was chosen for the selection of the PLC/PRF/5^Sor-R^ cells (Figure 1A). After the selection and recovery stages, Sorafenib-resistant PLC/PRF/5^Sor-R^ pool 1 was successfully propagated and maintained in media containing their IC_50_ Sorafenib concentration of 5.5 μM (Figure 1C).

### 3.2. Identification of Candidate Genes Involved with Drug-Resistance

Sorafenib-resistant HCC cell line PLC/PRF/5 was established (PLC/PRF/5^SorR^ mutant cell pool) and used to identify potential candidate genes for drug resistance by analyzing the differentially expressed gene (DEG) profiling by RNA-seq. In summary, a total of 16,442 DEGs were identified in PLC/PRF/5^SorR^ mutant pool when compared with its wild-type cells (Appendix A). From these DEGs, 13,187 DEGs were unchanged between PLC/PRF/5^SorR^ mutant pool and the wild-type cells (Appendix A). When the remaining 3255 DEGs identified in the PLC/PRF/5^SorR^ mutant pool were compared with its wild-type cells, 2240 genes were upregulated and 1015 genes were downregulated (Appendix A). Under further stringent selection criteria (*p*-value < 0.01, FDR < 0.02, Fold-change: ±1.5), 142 candidate genes were identified as contributing to Sorafenib resistance, with 96 upregulated genes and 46 downregulated genes (Appendix A). Subsequently, Kyoto Encyclopedia of Genes and Genomes (KEGG) analysis using DAVID bioinformatics resources on these 142 DEGs from the PLC/PRF/5^Sor-R^ resistant cells predicted the PI3K/AKT signaling pathway as the top enriched pathway (*p*-value = 4.08 × 10^−6^) contributing to drug resistance (Table 1). In addition, IL7 signaling pathway was also enriched and predicted to be activated in canonical pathway enrichment analysis by Ingenuity Pathways Analysis (Figure 1E).

### 3.3. Clinical Relevance of IL7 and MAL2 in HCC-Associated Drug Resistance

First, *IL7* was identified as being part of the PI3K/AKT signaling pathway (Table 1) as well as one of the top ten upregulated genes in PLC/PRF/5^Sor-R^ resistant cells (Table 2).

The top ten upregulated genes were then investigated using the TCGA liver cancer database for their clinical relevance (*n* = 360) (Appendix A). Interestingly, a significant tendency for co-occurrence between *IL7* and *MAL2* was only found amongst the top ten upregulated genes in the TCGA liver cancer database (*p* < 0.001). *IL7* and *MAL2* were found to be dysregulated in 21% and 30% of HCC patients, respectively (Appendix A). Both *IL7* and *MAL2* were confirmed to be significantly overexpressed in PLC/PRF/5^SorR^ mutant cell pool compared to the wild-type parental cell line; *IL7* was significant upregulated by +114-fold (*p*-value 8.29 × 10^−12^), while *MAL2* was significant upregulated by +201-fold (*p*-value 7.36 × 10^−28^) (Appendix A). Both genes were either amplified or showed high mRNA expression in the majority of affected HCC patients, indicating a gain-of-function phenotype. In order to validate our candidate genes translationally, the online database Gene Expression Omnibus (GEO) was used for comparison [28]. Importantly, significant upregulated expression levels of *IL7* and *MAL2* were observed in Sorafenib-resistant HCC patients using the GEO database GSE109211 (Appendix A) [29]. Moreover, positive correlation between *IL7* and *MAL2* was also found in both GSE109211 and TCGA (Appendix A). Taken together, *IL7* and *MAL2* can potentially co-occur in the contribution to Sorafenib resistance in HCC. In corroboration with the RNA-seq data, both *IL7* and *MAL2* were confirmed to be significantly overexpressed in the PLC/PRF/5^SorR^ mutant cell pool compared to the wild-type parental cell line by qPCR (Figure 2A). In addition, these Sorafenib-resistant mutant cells had significantly greater colony forming ability when compared with their parental cells under different Sorafenib concentrations (Figure 2B).

### 3.4. In Vitro Validation of IL7 and MAL2 in HCC-Associated Drug Resistance

The piggyBac (*PB*)-based transposon vector system was used as a gene delivery tool to stably overexpress *IL7* and/or *MAL2* in parental PLC/PRF/5 cells to determine their effect on Sorafenib resistance by two parameters: colony formation ability and anti-apoptotic effect. Overexpression of *IL7* and/or *MAL2* were confirmed in transfected PLC/PRF/5 cells by qPCR (Figure 2C). Interestingly, clonogenic survival assay results revealed that both *IL7* and *MAL2* were required to induce Sorafenib resistance in PLC/PRF/5 cells, while overexpressing either *IL7* or *MAL2* alone does not significantly induce Sorafenib resistance (Figure 2C). Using an apoptosis assay, PLC/PRF/5 cells that overexpressed *IL7* and/or *MAL2* were more anti-apoptotic compared with wild-type or *GFP*-transfected cells when treated with Sorafenib (Figure 2D,E).

### 3.5. Validating the Role of IL7 and MAL2 in HCC Drug Resistance by Dysregulating JAK/STAT and PI3K/AKT Signaling Pathways

The binding of IL7 to IL7 receptors on the cell membrane activates JAK/STAT and PI3K/AKT downstream signaling pathways favors both proliferation and anti-apoptosis [30,31,32]. In addition, since MAL2 can facilitate IL7 to traffic via endosomal compartments to plasma membrane for IL7 receptor binding [33,34]. These previous observations strongly support our hypothesis that both *IL7* and *MAL2* play essential roles in HCC drug resistance by dysregulating JAK/STAT and PI3K/AKT signaling pathways. Interestingly, RNA-seq results from the PLC/PRF/5 Sorafenib-resistant mutant pool also corroborated this hypothesis: the upregulation of genes associated with the JAK/STAT signaling pathway, such as *JAK3* (+2.89-fold, *p*-value = 2.05 × 10^−11^), *STAT1* (+2.53-fold, *p*-value = 0.00) and *STAT2* (+2.17-fold, *p*-value = 8.04 × 10^−78^) (Appendix A); and the upregulation of genes associated with the PI3K/AKT signaling pathway, such as phosphoinositide-3-kinase interacting protein 1 (*PIK3IP1*, +4.14-fold, *p*-value = 1.63 × 10^−7^) and phosphoinositide-3-kinase regulatory subunit 3 (*PIK3R3*, +3.27-fold, *p*-value = 9.75 × 10^−50^) and *SFN* (+2.95-fold, *p*-value = 7.31 × 10^−171^) (Appendix A). Our RNA-seq data showed genes of the JAK/STAT and PI3K/AKT signaling pathways were upregulated, such as interleukin 2 receptor subunit gamma (*IL2RG*, +61.4-fold, *p*-value = 5.04 × 10^−17^) and *CDKN1A* (*p*-value +6.73-fold, *p*-value = 2.46 × 10^−97^) (Appendix A). Upregulation of both downstream effector genes, *CDKN1A* and *SFN*, contributed to the anti-apoptotic effect seen in Sorafenib-resistant PLC/PRF/5 cells by dysregulating the SAPK/JNK signaling pathway (Appendix A). In corroboration with the RNA-seq results, Western blot results showed STAT3 activation in *IL7* and *MAL2* overexpressing PLC/PRF/5 cells compared with GFP control PLC/PRF/5 cells (Figure 3A).

Furthermore, higher AKT and PI3K activities were also detected in overexpressing-*IL7* and/or *MAL2* PLC/PRF/5 cells by Western blot analyses (Figure 3B). STAT3 and PI3K/AKT downstream effector *CDKN1A* were significantly induced in *IL7* and/or *MAL2* overexpressing PLC/PRF/5 cells under Sorafenib treatment by qPCR (Figure 3C). Preliminary Western blot results showed reduced phosphorylated JNK (p-JNK) levels in *IL7* and *MAL2* overexpressing PLC/PRF/5 cells compared with GFP control PLC/PRF/5 cells under Sorafenib treatment in a dose dependent manner (Figure 3D).

### 3.6. Autophagy-Inducing Stapled Peptides Readily Enhanced Autophagic Flux in Both Wild-Type and Sorafenib-Resistant HCC Cells Overexpressing IL7 and MAL2

With *IL7* and *MAL2* validated as two candidate genes contributing to Sorafenib resistance, we searched for pharmacological intervention that could overcome such resistance. In our previous studies, we developed a series of autophagy-inducing stapled peptides by targeting Beclin 1, an essential autophagy protein that serves as a scaffolding member of the Class III PI3K complex, to promote early-stage autophagy induction and late-stage autophagosome-lysosome fusion [24,25,35]. These Beclin 1-targeting stapled peptides readily induced autophagy and promoted endolysosomal degradation of oncogenic receptors that are frequently overexpressed on plasma membrane, like erb-b2 receptor tyrosine kinase 2 (ERBB2) and MET [24,25,36,37,38]. Additionally, these stapled peptides also exerted potent anti-proliferative effect on cancer cells by inducing non-apoptotic cell death [25,36,37]. In particular, these autophagy-inducing stapled peptides promoted the endolysosomal degradation of MET oncogene and overcame adaptive Sorafenib resistance in MET^+^ HCC cells [37]. Given the critical role of *MAL2* in endocytic trafficking and the crosstalk of this process with autophagy, we reason that our designed peptides may trigger autophagy and promote endolysosomal degradation to overcome *IL7*/*MAL2*-mediated Sorafenib resistance.

To test this hypothesis, we first assessed the basal autophagy activity in PLC/PRF/5 cells transfected with either GFP alone (GFP), to represent the Sorafenib-sensitive type, or overexpressing both IL7 and MAL2 (IL&MAL2) to represent the Sorafenib-resistant type. Treatment by Rapamycin, a known autophagy inducer, failed to trigger an autophagic response in GFP cells as the autophagy markers LC3-II and p62 showed little change (Figure 4A,B). In contrast, Rapamycin treatment led to significant increase in LC3-II level in IL7&MAL2 cells, thus suggesting that MAL2 over-expression may enhance both endocytic trafficking and autophagy (Figure 4A,B).

We then treated the GFP and IL7&MAL2 cells with Tat-SP4 and Tat-SP9, two autophagy-inducing stapled peptides developed in our previous studies [24,25]. Both peptides target the Beclin 1 coiled coil domain to induce autophagy, but Tat-SP9 showed higher binding affinity to Beclin 1 than Tat-SP4 [24,25]. Additionally, Tat-SP9 also showed more potent anti-proliferative efficacy than Tat-SP4 in ERBB2-POSITIVE breast cancer cells [25]. Here, our results show that both Tat-SP4 and Tat-SP9 readily induced autophagy in GFP cells as the level of LC3-II was noticeably increased in dosage-dependent manner, as well as in the absence of presence of chloroquine (CQ), an autophagy inhibitor (Figure 4C,D). Similar results were observed for IL7&MAL2 cells (Figure 4E,F), thus suggesting that our designed peptides could induce autophagy in HCC cells regardless of the IL7 and MAL2 levels. In summary, our study shows that overexpression of IL7 and MAL2 moderately enhanced the basal autophagic response in HCC cells, but this change does not affect the autophagy-inducing activity of our designed peptides.

### 3.7. Autophagy-Inducing Stapled Peptide Showed Synergic Toxicity with Sorafenib in Drug-Resistant IL7&MAL2 Cells

To assess whether our autophagy-inducing stapled peptides could overcome Sorafenib resistance, we treated both GFP and IL7&MAL2 cells with Tat-SP4 and Tat-SP9 and measured cell viability using the trypan blue exclusion assay. Our results show Tat-SP4 exerted mild anti-proliferative effect on both GFP and IL7&MAL2 cells with IC_50_ of ~69.71 μM and 69.89 μM respectively (Figure 4G). In comparison, Tat-SP9 showed much higher potency on these two cell lines with IC_50_ of ~12.63 μM and 12.85 μM respectively (Figure 4G).

We also used clonogenic survival assay to further assess the anti-proliferative efficacy of our designed peptides in GFP and IL7&MAL2 cells. We decided to focus on Tat-SP9 only given its better potency than Tat-SP4 as measured by the Trypan Blue assay. Our clonogenic data shows that Tat-SP9 significantly reduced the colony numbers of both GFP and IL7&MAL2 cells in a dosage-dependent manner, further corroborating the IC_50_ values obtained from the Trypan Blue assay (Figure 4H). Additionally, while IL7&MAL2 cells were resistant to Sorafenib, co-treatment with Tat-SP9 overcame such resistance and led to synthetic lethality as no colonies were detected under the co-treatment conditions (Figure 4I). This data suggests the exciting possibility that Tat-SP9 may override IL7&MAL2-mediated Sorafenib resistance in HCC cells.

## 4. Discussion

In the current study, *SB* insertional mutagenesis was used in vitro to generate Sorafenib-resistant HCC cells. This method of generating drug-resistant cell lines allows for an effective way to generate various resistant cell line models rapidly by high dosage drug selection after *SB* transposon mutagenesis (Figure 1). Genetic analysis by RNA-seq identified *IL7* and *MAL2* in contributing to PLC/PRF/5^Sor-R^ resistant cells. Clinically, *IL7* and *MAL2* were found to be consistently dysregulated in both HCC and Sorafenib-resistant HCC patients using TCGA and GEO databases, respectively. In addition, there is significant tendency for co-occurrence between *IL7* and *MAL2*. Compensatory activation of different survival pathways in Sorafenib treatment has been shown to occur as a mechanism for chemoresistance [39]. The role of *IL7* and *MAL2* in contributing to Sorafenib resistance were validated by overexpressing experiments in parental PLC/PRF/5 cells and observing for colony formation ability and anti-apoptotic effect. Interestingly, both *IL7* and *MAL2* were required to induce Sorafenib resistance in PLC/PRF/5 cells, while overexpressing either gene alone does not significantly induce Sorafenib resistance (Figure 2C). Cells that overexpress *IL7* and/or *MAL2* were more anti-apoptotic compared with wild-type or *GFP*-transfected cells when treated with Sorafenib (Figure 2D,E). IL7 exerts its biological functions primarily by activating the IL7 receptor (IL7R) to mediate various signaling pathways and shape tumor immunity [30]. IL7 has been reported to exhibit a dual role in cancer pathophysiology [30]. Recent studies have revealed that IL7 plays a significant role in glioma, melanoma and leukemia by contributing to anti-tumor effects through the release of cytokines, including IFNG, IL1A, IL1B, TNF, IL2 and IL4 [40,41,42]. In contrast, IL7 has been reported to promote the proliferation and survival of tumor cells by activating the JAK/STAT, PI3K/AKT and RAS/ERK signaling pathways in non-small cell lung cancer, bladder cancer and T-cell acute lymphoblastic leukemia [15,42,43]. MAL2 is part of the tetraspanin family of membrane structural proteins, which play essential roles in both raft-associated membrane trafficking during transcytosis and vesicle transport [17,44]. Recent research has reported that the aggregation of smaller rafts into larger complexes can serve as platforms for signal transduction of receptor tyrosine kinase (RTK) signaling by harboring receptors and regulatory molecules in cancers [45,46]. MAL2 exhibits high expression levels in diverse cancers and is closely linked to cancer development and prognosis, such as breast cancer, ovarian cancer and pancreatic cancer [33,47,48]. In liver cancer, a study has shown that MAL2 is associated with early stage HCC rather than late stage [49]. Additionally, MAL2 has also been reported to enhance resistance to trastuzumab in breast cancer cells by stabilizing HER2 [45]. Nevertheless, the mechanistic studies on the role of both IL7 and MAL2 in HCC-associated drug resistance remain limited.

It is hypothesized that the binding of self-secreted IL7 and its cytokine receptors leads to the activation of JAK/STAT and PI3K/AKT signaling pathways to sustain HCC cellular proliferation and survival during Sorafenib treatment [13,15,16]. Our Western blot analyses confirmed STAT3 activation in *IL7* and *MAL2* overexpressing PLC/PRF/5 cells compared with GFP control PLC/PRF/5 cells (Figure 3A). Activation of the PI3K/AKT signaling pathways to sustain HCC cellular proliferation and survival was also confirmed by Western blot analyses in *IL7* and *MAL2*-overexpressing PLC/PRF/5 cells (Figure 3B). In addition, cyclin dependent kinase inhibitor 1A (CDKN1A) expression was significantly induced in *IL7* and/or *MAL2* overexpressing PLC/PRF/5 cells under Sorafenib treatment (Figure 3C). A previous study reported that CDKN1A can potentially serve as a prognostic signature of HCC resistance to Sorafenib [50]. Moreover, CDKN1A is a downstream effector that contributes to the anti-apoptotic effect via inhibiting apoptosis-induction protein, such as apoptosis-related caspases, stress-activated protein kinases (SAPK)/Jun amino-terminal kinases (JNK) and apoptosis signal-regulating kinase 1 (ASK1) [51,52]. Another study has shown that the activation of JNK can augment the sensitivity of HCC to Sorafenib [53]. In addition, JNK-mediated apoptosis was also reported to play a role in augmenting HCC cell death triggered by Sorafenib [54]. Our Western blot results also showed similar results that the activation of JNK was induced by Sorafenib in GFP control of PLC/PRF/5 cells (Figure 3D). Interestingly, Western blot results showed reduced phosphorylated JNK (pJNK) levels in *IL7* and *MAL2*-overexpressing PLC/PRF/5 cells compared with GFP control PLC/PRF/5 cells under Sorafenib treatment in a dose dependent manner (Figure 3D). The cascade of JNK-mediated apoptosis induced by Sorafenib was inhibited in *IL7* and *MAL2*-overexpressing PLC/PRF/5 cells. Finally, it is also hypothesized that MAL2, a transmembrane protein, contributes to intracellular trafficking and secretion of IL7 to the plasma membrane for the binding of cytokine receptors in order to trigger downstream signaling pathways that contributes to drug resistance (Figure 3E).

With IL7 and MAL2 identified as two candidate genes that contribute to Sorafenib resistance in HCC cells, we set out to search for pharmacological interventions that may override such resistance. In our search, we have turned to autophagy-inducing stapled peptides developed in our previous studies based on the following considerations. First, as an evolutionarily conserved process that degrades and recycles cytosolic content in lysosome-dependent manner, autophagy is intricately involved in cancer and acts as a double-edged sword playing both pro- and anti-survival roles with context-dependent manner [55,56]. Second, autophagy has extensive crosstalk with the endolysosomal degradation pathway because both involve endocytic trafficking of cargos to the lysosomes. Thus, the induction of autophagy induction may promote the endolysosomal degradation process to counter the endocytic trafficking of IL7 and its cognate receptors mediated by MAL2.

Our results demonstrated that two autophagy-inducing stapled peptides, Tat-SP4 and Tat-SP9, readily induced autophagy in both wild-type and Sorafenib-resistant HCC cells and inhibited their proliferation with comparable potency. Additionally, our stapled peptides showed synergy with Sorafenib in *IL7*/*MAL2*-overexpressing HCC cells to overcome drug resistance. These results further validate autophagy as a potential target for novel anti-cancer therapies, especially to overcome *IL7*/*MAL2*-mediated Sorafenib resistance.

Taken together, our in vitro screen identified novel genetic interactions that contribute to Sorafenib resistance in HCC and may play an important role in primary and acquired drug-resistances. Additionally, our autophagy-inducing peptides offer possible pharmacological strategies to overcome such *IL7*/*MAL2*-mediated drug resistance and enhance the therapeutic effect of existing drugs like Sorafenib in devastating diseases like HCC.

## 5. Conclusions

Transposon insertional mutagenesis was used to generate Sorafenib-resistant HCC cells and identified *IL7* and *MAL2* as candidate drug resistance genes. Their roles were validated in vitro and the resistance mechanism involved the activation of pro-survival JAK/STAT and PI3K/AKT signaling pathways. *IL7* and *MAL2* co-overexpressing cells also showed higher anti-apoptotic effect, clonogenic survival, and increased PI3K/AKT/STAT3 activities. In addition, autophagy inducing stapled peptides could induce autophagy and inhibited the proliferation of both wild-type and Sorafenib-resistant HCC cells that co-overexpress both *IL7* and *MAL2*. Furthermore, these peptides exhibited synergistic cytotoxicity with Sorafenib in drug-resistant HCC cells co-overexpressing *IL7* and *MAL2*. Therefore, targeting autophagy could serve as a promising novel approach to overcome IL7/MAL2-mediated Sorafenib resistance in HCC.

## Figures and Tables

**Figure 1 cancers-15-05280-f001:**
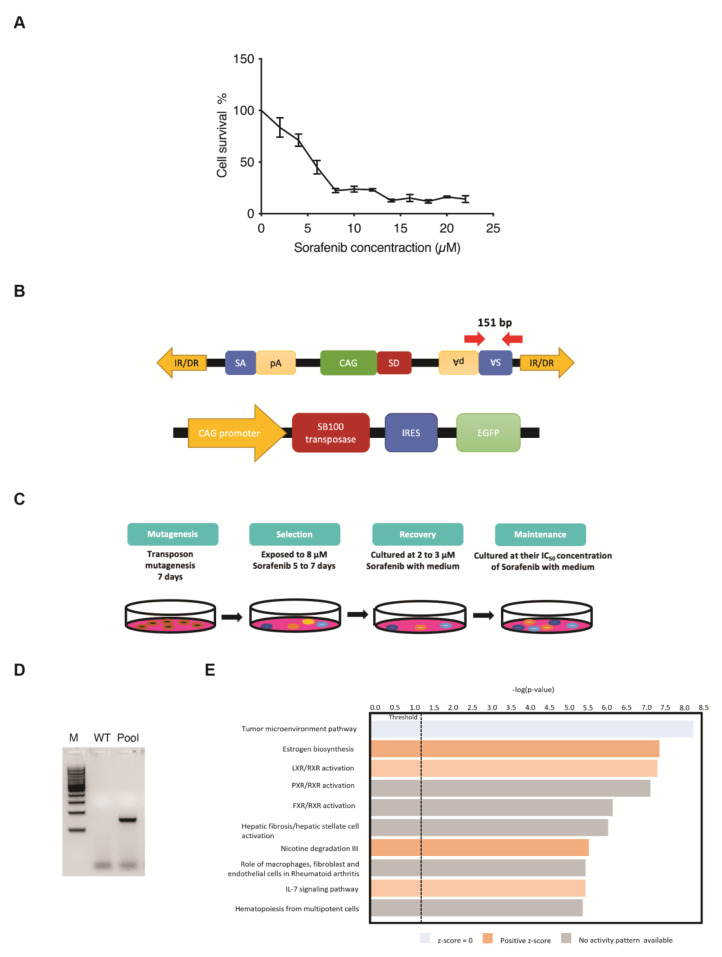
Generating drug-resistant human HCC cell lines using the Sleeping Beauty (*SB*) transposon insertional mutagenesis system. (**A**) Five-day kill curves using different concentrations of Sorafenib for IC_20_, IC_50_ and IC_90_ measurements. (**B**) Schematic diagram of the *SB* mutagenic transposon T2/Onc3 (**top**) and SB100 transposase (**bottom**) vectors used in the screen. CMV enhancer/chicken β-actin promoter (CAG) promoter with splice donor (SD) and splice acceptor (SA) with polyadenylation tails (pA) to drive upregulation of genes and/or downregulation due to premature truncation, respectively. *SB* transposon contains 210–250 bp inverted repeats (IRs) at their terminus and direct repeats DNA sequence motifs (DRs) at the ends of each IR. SB100 transposase recognizes the IR/DRs terminals of the transposon to excise and facilitate the insertion of *SB* transposon into chromosomal DNA. IRES, internal ribosome entry site; EGFP, enhanced green fluorescent protein. Red arrows indicate primer pair set used for amplifying the SA sequence of transposon and expected amplicon size. (**C**) Outline of the screening strategy for generating Sorafenib-resistant cell lines. Cells were transfected with T2/Onc3 and SB100 vectors to drive mutagenesis for seven days (mutagenesis stage). During the selection stage, Sorafenib concentration of 8 μM was used for PLC/PRF/5, until all parental cells were nearly dead (~1 week). Cells that survived the selection stage were treated with 2 μM Sorafenib for recovery to 80% confluence (recovery stage) and then maintained by culturing in the presence of Sorafenib at their parental IC_50_ concentration (maintenance stage). (**D**) Representative PCR demonstrating the presence of transposon insertions in transfected PLC5/PRF/5 cells by the successful amplification of the SA sequence with the expected 151 bp amplicon. WT, wild-type non-transfected cell line; Pool, mutant cell pool resistant to Sorafenib; M, 100 bp DNA ladder. (**E**) Canonical pathway enrichment analysis using IPA.

**Figure 2 cancers-15-05280-f002:**
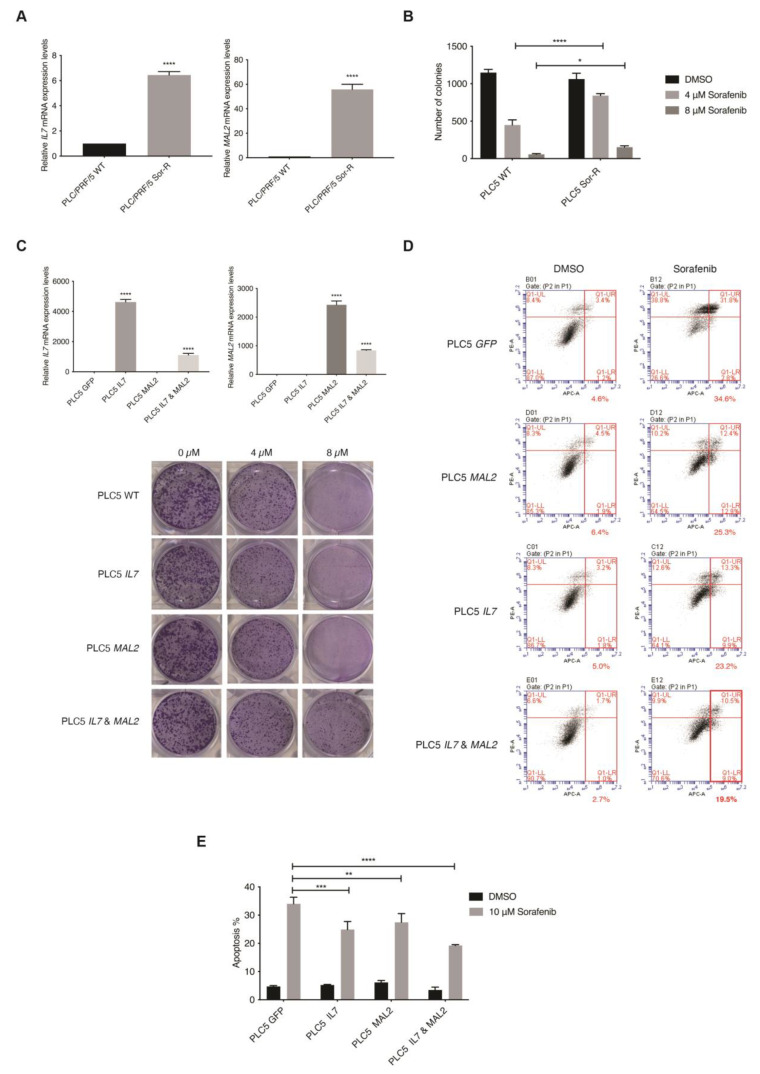
In vitro validation of the role of *IL7* and *MAL2* in HCC-associated Sorafenib drug resistance. (**A**) *IL7* and *MAL2* were significantly overexpressed in Sorafenib-resistant mutant PLC/PRF/5 cell pool (Sor-R) compared to the wild-type (WT) parental cell line by qPCR. ****, *p* < 0.0001; Student unpaired *t*-test. (**B**) Graphical representation showing significantly more colony forming ability by Sorafenib-resistant mutant PLC/PRF/5 (PLC5 Sor-R) cells when compared with wild-type (WT) parental PLC/PRF/5 (PLC5) cells under different Sorafenib concentrations (4 and 8 µM). ****, *p* < 0.0001; *, *p* < 0.05; two-way ANOVA test. (**C**) Overexpression of *IL7* and/or *MAL2* were confirmed in transfected PLC5 cells by qPCR compared with GFP control transfected cells. ****, *p* < 0.0001; Student unpaired *t*-test. Representative clonogenic survival assay images of PLC5 cells overexpressing *IL7* and/or *MAL2* under different Sorafenib concentrations. (**D**) Co-expression of *IL7* and *MAL2* in PLC5 cells treated with 10 µM Sorafenib induced greater anti-apoptotic effect compared with control *GFP*-overexpressing cells. Flow cytometry profiles represented intensity of Annexin-V-APC staining in *X*-axis and PI in *Y*-axis. (**E**) Graphical quantification of the apoptosis assay. ****, *p* < 0.0001; ***, *p* < 0.001; **, *p* < 0.01; two-way ANOVA test.

**Figure 3 cancers-15-05280-f003:**
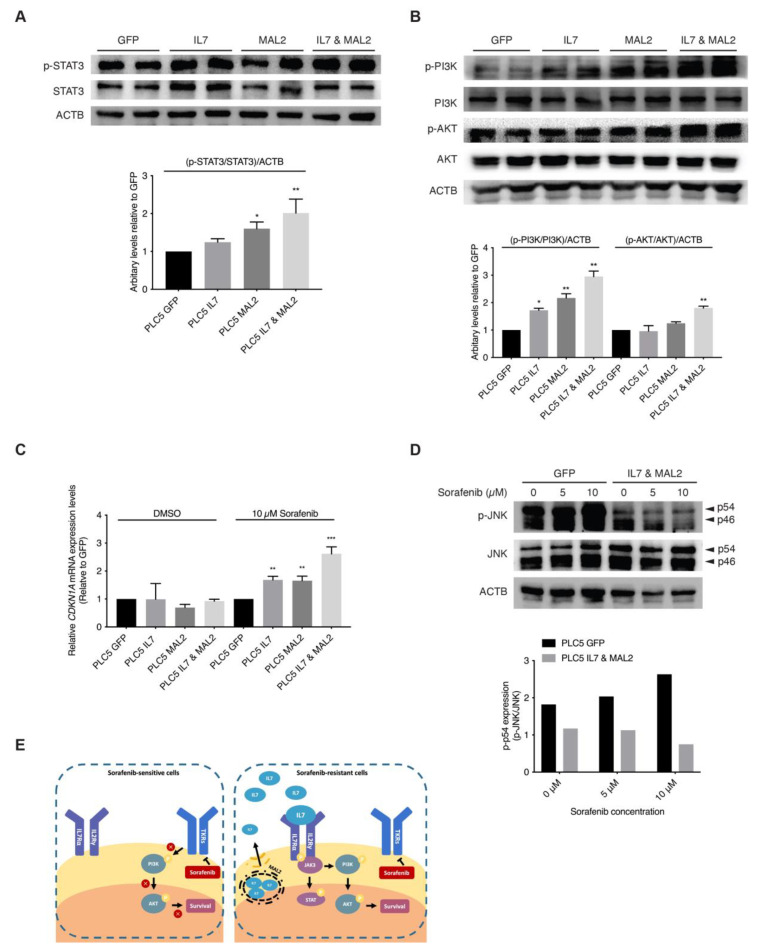
Activation of JAK/STAT and PI3K/AKT signaling pathways by IL7 and MAL2 in HCC-associated Sorafenib resistance. (**A**) Representative Western blot and relative protein level analyses confirming STAT3 activation in *IL7* and *MAL2* overexpressing PLC/PRF/5 cells compared with GFP control PLC/PRF/5 cells. *, *p* < 0.05; **, *p* < 0.01; two-way ANOVA test. (**B**) Representative Western blot and relative protein level analyses confirming PI3K/AKT activation in *IL7* and *MAL2* overexpressing PLC/PRF/5 cells compared with GFP control PLC/PRF/5 cells. *, *p* < 0.05; **, *p* < 0.01; two-way ANOVA test. (**C**) *CDKN1A* expression was significantly induced in *IL7* and/or *MAL2* overexpressing PLC/PRF/5 cells under Sorafenib treatment. **, *p* < 0.01; ***, *p* < 0.001; Student unpaired *t*-test. (**D**) Representative Western blot and relative protein level analyses showing reduced phosphorylated JNK (p-JNK) levels in *IL7* and *MAL2* overexpressing PLC/PRF/5 cells compared with GFP control cells under Sorafenib treatment in a dose dependent manner. (**E**) Summary diagram showing MAL2, a transmembrane protein, contributes to intracellular trafficking and secretion of IL7 to the plasma membrane. The binding of IL7 to cytokine receptors result in the activation of important downstream survival signaling pathways that contributes to Sorafenib resistance. The uncropped bolts are shown in Appendix A.

**Figure 4 cancers-15-05280-f004:**
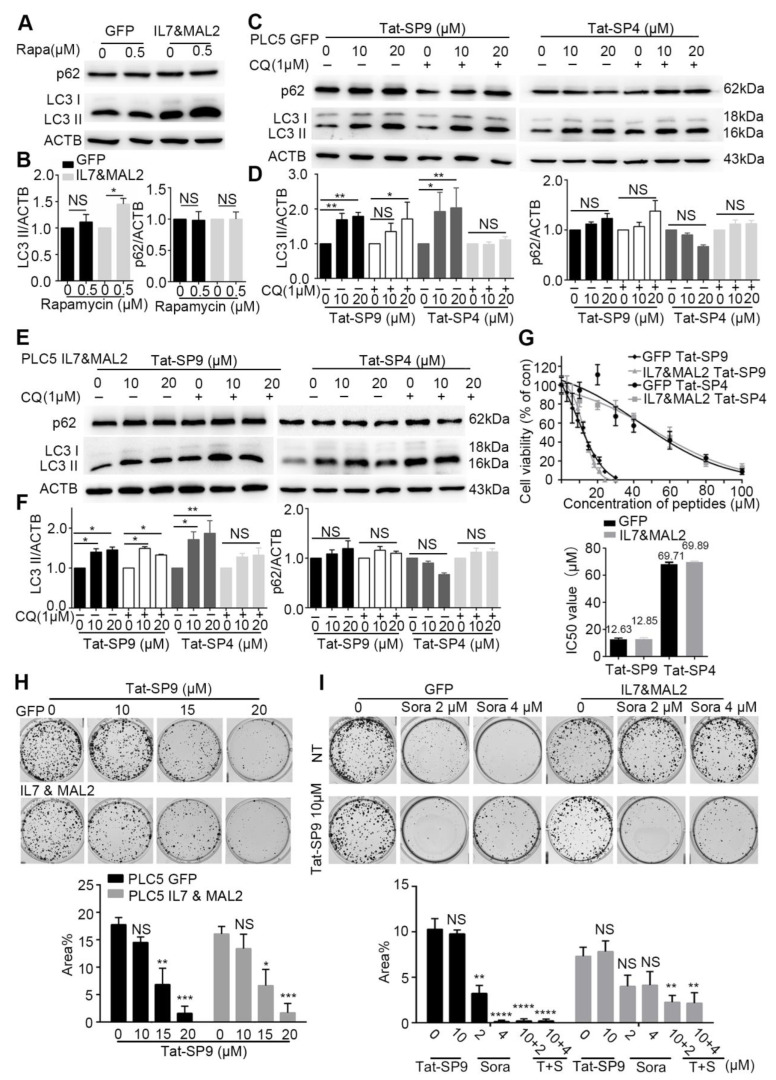
Autophagy-inducing stapled peptides exerted a synergistic anti-proliferative effect with Sorafenib in resistant HCC cells that co-overexpress both IL7 and MAL2. (**A**) Autophagy markers p62 and LC3-II were measured by Western blot in IL7 and MAL2-overexpressing PLC/PRF/5 cells compared with GFP control PLC/PRF/5 cells after treatment with or without the autophagy inducer rapamycin (500 nM, 3 h). (**B**) Statistic analysis of LC3-II and p62 level change of (**A**) after normalization to the level of β-Actin. (**C**) The two autophagy markers were assessed using Western blot in GFP control PLC/PRF/5 cells after treatment with Tat-SP9 (0, 10, 20 μM for 12 h) and Tat-SP4 (0, 10, 20 μM for 3 h) with or without 1 μM CQ. (**D**) Statistic analysis of LC3-II and p62 level change of (**C**) after normalization to the level of β-Actin. (**E**) Similar Western blots as (**C**), but cells are IL7 and MAL2-overexpressing PLC/PRF/5 cells. (**F**) Statistic analysis of LC3-II and p62 level change of (**E**) after normalization to the level of β-Actin. Bar represents mean ± SEM of three replicates. (**G**) Cell viability was assessed in *IL7* and *MAL2*-overexpressing PLC/PRF/5 cells and GFP control PLC/PRF/5 cells after treatment with different concentrations of Tat-SP9 and Tat-SP4. The estimated IC_50_ value for Tat-SP9 was 12.63 μM in GFP control PLC/PRF/5 cells and 12.58 μM in Sorafenib-resistant cells, while the estimated IC_50_ value for Tat-SP4 was 69.71 μM in control cells and 69.89 μM in resistant cells. (**H**) Clonogenic survival assay images were taken for both GFP control PLC/PRF/5 GFP cells and IL7 and MAL2-overexpressing PLC/PRF/5 cells under different concentrations of Tat-SP9 (10, 15, 20 μM). Bars represent mean ± SEM (n = 3). (**I**) Representative clonogenic survival assay images of GFP control PLC/PRF/5 cells and Sorafenib-resistant PLC/PRF/5 cells after treatment with Tat-SP9 (10 μM) or Sorafenib (2 or 4 μM), alone or combined. Bars represent mean ± SEM of three replicates. NS means no significant difference *, *p* < 0.05; **, *p* < 0.01; ***, *p* < 0.001,****, *p* < 0.0001 two-way ANOVA test. The uncropped bolts are shown in Appendix A.

**Table 1 cancers-15-05280-t001:** KEGG pathway enrichment analysis of PLC/PRF/5^Sor-R^ resistant cells.

Pathways	*p*-Value	Involved Genes
PI3K-Akt signaling pathway	4.08 × 10^−6^	*CDKN1A*, *G6PC*, *CSF1*, *PIK3R3*, *IL2RG*, *FGF2*, *NFKB1*, *NR4A1*, *COL3A1*, *KITLG*, *BCL2L11*, *IL7*, *SPP1*, *KDR*, *SGK1*, *PCK1*, *MET*
Steroid hormone biosynthesis	6.17 × 10^−5^	*UGT2B11*, *UGT1A1*, *AKR1C3*, *CYP1A1*, *AKR1C4*, *CYP3A5*, *CYP3A7*
Complement and coagulation cascades	7.72 × 10^−4^	*CPB2*, *SERPIND1*, *F12*, *C5AR1*, *KNG1*, *SERPINA5*, *C2*
Pathways in cancer	0.001003	*CEBPA*, *CDKN1A*, *CXCL8*, *CXCR4*, *PIK3R3*, *TGFA*, *PTGS2*, *FGF2*, *FOXO1*, *NFKB1*, *BMP2*, *KITLG*, *CXCL12*, *BIRC5*, *MET*, *WNT4*
Transcriptional misregulation in cancer	0.001217	*CEBPA*, *CDKN1A*, *CXCL8*, *IGFBP3*, *TMPRSS2*, *HMGA2*, *PROM1*, *MET*, *FOXO1*, *NFKB1*
Chemical carcinogenesis	0.00139	*UGT2B11*, *UGT1A1*, *CHRNA7*, *CYP1A1*, *PTGS2*, *CYP3A5*, *CYP3A7*
NF-kappa B signaling pathway	0.001556	*CXCL12*, *TICAM2*, *CXCL8*, *LY96*, *PTGS2*, *NFKB1*, *TNFSF13B*
Arachidonic acid metabolism	0.001901	*CYP2J2*, *GPX2*, *GPX3*, *AKR1C3*, *GGT1*, *PTGS2*
Retinol metabolism	0.002199	*CYP26B1*, *UGT2B11*, *UGT1A1*, *CYP1A1*, *CYP3A5*, *CYP3A7*
Cytokine-cytokine receptor interaction	0.002682	*IL22RA1*, *BMP2*, *CXCL12*, *CXCL8*, *THPO*, *CSF1*, *IL7*, *CXCR4*, *IL2RG*, *CXCL5*, *TNFSF13B*

**Table 2 cancers-15-05280-t002:** Top ten upregulated genes in PLC/PRF/5^Sor-R^ resistant cells.

Altered Genes	Log2 Fold-Change	*p*-Value
*EPPIN*	10.920	2.99 × 10^−182^
*PTGS2*	9.926	1.33 × 10^−212^
*ALB*	9.351	0
*KDR*	8.758	5.76 × 10^−128^
*FABP1*	8.011	3.88 × 10^−7^
*MAL2*	7.651	7.36 × 10^−28^
*ABCG1*	7.082	2.02 × 10^−37^
*IL7*	6.833	8.29 × 10^−12^
*IQGAP2*	6.755	1.38 × 10^−152^
*SLCO3A1*	6.219	9.21 × 10^−27^

## Data Availability

The data presented in this study are available in the paper. Additional data related to this paper are available on request from the corresponding author.

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
