# Peer review of "Sorafenib Resistance Contributed by IL7 and MAL2 in Hepatocellular Carcinoma Can Be Overcome by Autophagy-Inducing Stapled Peptides"

_cancers, 2023, doi:10.3390/cancers15215280_

Round 1

Reviewer 1 Report (Previous Reviewer 1)

Comments and Suggestions for Authors

The authors have addressed all my comments.

Author Response

Reviewer 2 Report (Previous Reviewer 2)

Comments and Suggestions for Authors

I have no more comments. 

Author Response

This manuscript is a resubmission of an earlier submission. The following is a list of the peer review reports and author responses from that submission.

Round 1

Reviewer 1 Report

Comments and Suggestions for Authors

This manuscript identified the importance of IL7 and MAL2 in developing sorafenib resistance relevant to HCC disease. Further, the authors show the signaling pathway involved in resistance. Moreover, the authors identified that autophagy induction could overturn sorafenib resistance. However, the authors did not mechanistically prove the involvement of autophagy. The authors should provide more information about the experiments and use proper controls. Also, the authors should re-do the statistics in all the figures, as unpaired student t-tests should not be performed for more than 2 samples in a group. 

Major Comments

·      In lines 19-22, the authors did not show any evidence of endocytosis in the manuscript.

·      The authors should clarify to the readers how the SB transposon system helps generate sorafenib-resistant cells in the results section.

·      In lines 302-304, the authors should show the IL7 pathway enrichment data.

·      How did the authors verify the over-expression of IL-7 and MAL2? Why do the control cells express an unrelated protein, GFP, whereas the IL-7 and MAL2 over-expressing cells do not express GFP as fusion? These cells are not comparable.

·      In Fig. 3, the authors should double normalize the proteins relative to the loading control and plot as (p-STAT3/STAT3)/ACTB. The authors should do an ANOVA test to find the significance, and an unpaired t-test is inappropriate. Why western blotting was not performed to check CDKN1A levels? Also, some blots are saturated; the authors should re-quantify them with non-saturated blots. How many replicates were performed in panel D? 

·      In Figure 4, the author shows the effect of autophagy-inducing peptides on sorafenib-resistant cells. However, the authors should show how autophagy affects sorafenib-resistant cellular survival. What happens to the IL7 and MAL2 proteins upon autophagy induction? Why is LC3 I not induced upon rapamycin treatment? The authors should show the LC3II: LC3I ratio normalized to loading control.

·      Have the authors investigated p-STAT3, p-AKT, and p-PI3-K levels in the sorafenib-resistant cells treated with autophagy-inducing agents? The authors should include IL-7 or Mal2 (individually) over-expressing cells in Fig. 4H and I.

Minor Comments

·      Add citation for Sleeping Beauty (SB) transposon insertional mutagenesis in the introduction section

Comments on the Quality of English Language

Language usage is fine.

Reviewer 2 Report

Comments and Suggestions for Authors

1. Consider revising the title to make it more concise and informative. For example: "IL7 and MAL2 Contribute to Sorafenib Resistance in Hepatocellular Carcinoma: Overcoming Resistance with Autophagy-Inducing Stapled Peptides.

2. The abstract provides a brief overview of the study. It covers the background, methods, and key findings. However, it would be helpful to include the objectives of the study as well.

3. Instead of stating "Our studies suggest," it would be stronger to present the findings directly.

4. Consider adding more specific information about the role of IL7 and MAL2 in HCC and their potential contributions to drug resistance.

5. Provide information on the cell lines used, transfection methods, drug treatment protocols, and statistical analysis.

6. Consider addressing the limitations of the study and suggesting future research directions.

7. Provide additional information on the methods used for the synthesis, purification, and characterization of Tat-SP4 and Tat-SP9. 

Cite this reference for the synthesis of anti-angiogenesis receptor tyrosine kinase inhibitors. This review also introduces regorafenib as an analog of sorafenib. https://link.springer.com/article/10.1007/s11030-022-10406-8

Comments on the Quality of English Language

 Minor editing of the English language is required.